# Encapsulation of *Ammoides pusila* Essential Oil into Mesoporous Silica Particles for the Enhancement of Their Activity against *Fusarium avenaceum* and Its Enniatins Production

**DOI:** 10.3390/molecules28073194

**Published:** 2023-04-03

**Authors:** Yasmine Chakroun, Youssef Snoussi, Mohamed M. Chehimi, Manef Abderrabba, Jean-Michel Savoie, Souheib Oueslati

**Affiliations:** 1INRAE, UR1264 MycSA, CS2032, 33882 Villenave d’Ornon, France; 2IPEST, Laboratory Molecules Materials and Applications (LMMA), University of Carthage, La Marsa, Tunis 2070, Tunisia; 3CNRS, UMR 7182 ICMPE, 2-8 Rue Henri Dunant, 94320 Thiais, France; 4ITODYS, UMR 7086, Université Paris Cité & CNRS, 75013 Paris, France

**Keywords:** nanoencapsulation, essential oil, antifungal activity, *Fusarium*, enniatins, mycotoxins, chitosan

## Abstract

Essential oils (EOs) that have antifungal activity and mycotoxin reduction ability are candidates to develop bioactive alternatives and environmentally friendly treatment against Fusarium species in cereals. However, their practical use is facing limitations such as high volatility, UV sensitivity, and fast oxidation. Encapsulation techniques are supposed to provide protection to the EOs and control their release into the environment. *Ammoides pusilla* essential oil (AP-EO) proved to be an efficient inhibitor of *Fusarium avenaceum* growth and its enniatins (ENNs) production. In the present work, AP-EO was encapsulated, using the impregnation method, into mesoporous silica particles (MSPs) with narrow slit pores (average diameter = 3.1 nm) and coated with chitosan. In contact assays using an agar medium, the antifungal activity of AP-EO at 0.1 µL mL^−1^ improved by three times when encapsulated into MSPs without chitosan and the ENNs production was significantly inhibited both in coated and non-coated MSPs. Controls of MSPs also inhibited the ENNs production without affecting the mycelial growth. In fumigation experiments assessing the activity of the EO volatile compounds, encapsulation into MSPs improved significantly both the antifungal activity and ENNs inhibition. Moreover, coating with chitosan stopped the release of EO. Thus, encapsulation of an EO into MSPs improving its antifungal and antimycotoxin properties is a promising tool for the formulation of a natural fungicide that could be used in the agriculture or food industry to protect plant or food products from the contamination by toxigenic fungi such as *Fusarium* sp. and their potential mycotoxins.

## 1. Introduction

Mycotoxins may be defined as toxic secondary metabolites produced by several fungal genera. They may accumulate in food and feed leading to massive losses and may cause health issues for humans and animals. In fact, they specifically affect cereal production worldwide, which is an issue because high-quality grain is a critical component to global food security [1].

*Fusarium* species, for instance, are responsible of the production of various groups of mycotoxins in cereals, with different chemical structures and properties. *Fusarium* species such as *F. gramineareum, F. culmorum*, *F. proliferatum*, and *F. verticillioides* are able to produce the major mycotoxins including Trichothecenes B (represented mainly by nivalenol (NIV), deoxynivalenol (DON), and its acetylated derivatives 3 and 15 ADON), as well as zearalenone and fumonisins. These species are also pathogenic agents of the *Fusarium* Head Blight (FHB), an ear disease of particular concern due to the production losses in addition to the mycotoxins that may accumulate into the harvested grains. The trio *F. graminearum*, *F. culmorum*, and *F. avenaceum* were predominantly associated with FHB in the Mediterranean region [2]. The latter is a member of the *Fusarium tricinctum* complex, in addition to *F. tricinctum* and *F. acuminatum*, which are less frequently encountered in the Mediterranean region and are not identified as virulent pathogenic agents for cereal spikes but can cause damping-off [3]. They may cause root and stem rot in cereals and various host species and are showing an increasing incidence in many cultivation areas worldwide [4,5,6]. Furthermore, this species complex may produce emergent mycotoxins including enniatins (ENNs), fusaproliferin, beauvericin, and moniliformin [7]. ENNs, mainly enniatin B (ENNB), are common grain contaminants worldwide, and reports on their frequent occurrence in human urine indicate a high incidence of human exposure [8]. These cyclodepsipeptides are ionospheric, which is their primary mechanism of action, and may also cause mitochondrial dysfunction, lysosomal alteration, cell cycle disruption, and lipid peroxidation. ENNs are also involved in oxidative stress and also exert cytotoxic activities [8]. Furthermore, ENNs are also known as *F. avenaceum* plant pathogenic contributors and its competition with other *Fusarium* species.

Global actions for mycotoxin management in cereals that include the use of synthetic fungicides are unfortunately not able to stop severe attacks [1], and ecofriendly solutions are being investigated to limit mycotoxin accumulation in food matrices. One of the promising areas of study is the use of EOs. These products are extracted from various aromatic plants and have been widely studied for their antifungal activity and for their impact on trichothecenes B biosynthesis [9,10,11,12]; however, they have been studied at a lower level for their potential to control the emerging mycotoxins such as ENNs and their producing *Fusarium* species [13]. *Ammoides pusilla* EO (AP-EO) was investigated in a previous study for its bioactivity towards *F. acuminatum* and *F*. *avenaceum*. It exerted significant inhibition potential on mycelial growth and ENNs production [13,14].

EOs are environmentally safe, biodegradable, volatile, and efficient at low levels [15]. Despite their proven preservative potential, their applications and formulations in real food systems remain a challenging task because of major disadvantages such as intrinsic volatility, poor solubility in aqueous system, and chemical instability when facing abiotic factors such as light, temperature variation, and humidity [16,17].

Developing alternatives to synthetic fungicide treatments using EOs relies on their stabilization, their protection, and the control of their release. Nanoencapsulation is one of the most promising possibilities to achieve these objectives [18]. Specifically, nanoencapsulation techniques are based on synthesis, characterization, and application of nano-sized delivery systems with at least one dimension less than 100 nm. Nanoencapsulation also exhibits the advantages of providing a large surface-to-volume area, mass transfer behavior, and precise EOs release kinetics and their bioactive compounds into the targeted food system. A wide range of materials can be used in nanoformulas depending on the desired applications [16].

Mesoporous silica particles (MSPs) have been previously used for EO encapsulation [19]. The particles provided a high surface-to-volume ratio, were biocompatible, and could be functionalized with a variety of moieties in different parts of the particle. The introduction of functional groups made the MSPs highly versatile and enabled their use in various applications [20,21,22,23,24]. Furthermore, the surface of the MSPs increased the solubility of EO in aqueous phase by reducing its hydrophobicity, allowing the EO to infiltrate fungal cells, causing loss of membrane integrity, and finally destroying the whole cells [24].

Alternatively, chitosan encapsulation has been also widely investigated. It has been considered one of the most valuable wall materials for the encapsulation of bioactive compounds such as EOs due to its characteristic attributes of abundance (the second most abundant polysaccharide in nature), biocompatibility, hydrophilicity, biodegradability, non-toxicity (GRAS status), and excellent film-, gel- and particle-forming properties coupled with interesting antifungal activity [16,24,25,26,27,28].

In the present study, an interest was given to AP-EO in order to assess the improvement of their antifungal potential towards *F. avenaceum* and its ENNs production by their encapsulation into MSPs and coating the resulting products with a chitosan layer. This strategy is original, has high potential, and has not been reported so far; thus, it is what worth being reported.

## 2. Results

### 2.1. Mesoporous Silica Nanoparticles Characterization

According to the IUPAC classification, the nitrogen adsorption–desorption isotherm of the MSPs (Figure 1A) belongs to type IV and the high hysteresis loop can be affiliated to the type H_4_, which suggests that the obtained powder had a mesoporous structure with narrow slit pores as defined by Sing et al. [29]. The pore size distribution (PSD) of MSPs indicated a narrow multimodal distribution with an average pore diameter of 3.1 nm (Figure 1B). The value of the specific surface area (SSA), evaluated by the BET method, was 487 m^2^ g^−1^.

The surface chemical composition was determined using XPS. Figure 2 shows XP spectra of the mesoporous silica. Survey region (Figure 2a) clearly shows a quasi-neat silica, similar to calcined mesoporous silica [30]. Interestingly, CTAB has been removed after synthesis as the Br3d and N1s regions are flat and noisy (Figure 2a). However, the C1s fitted spectrum (Figure 2b) shows substantial C-O type carbon atoms (C1s peak component at 286.9 eV), probably due to traces of P123. The other components are due to adventitious contamination. The apparent elemental composition determined by XPS is as follows: O, 61.3%; Si, 33.6%; C, ~5.1%. The O/Si atomic ratio is 1.82, which is close to the theoretical value of 2.

The thermogravimetric curve of the MSPs (Figure 3) exhibited a first weight loss at 100 °C corresponding most likely to the loss of physisorbed water and carbon dioxide. Then, a second weight loss was registered between 100 and 400 °C, which could be assigned to the thermal decomposition of the surfactants (CTAB, P123) resisting the washing step and the organic compounds remaining in the porous structure. The thermal analysis of the AP-EO/MSPs revealed a more important total weight loss level (around 65%) compared to 20% with the bare MSPs. The difference is mainly explained by the loss of the loaded EO emulsion (around 45%). Furthermore, the important weight loss observed between 100 and 200 °C could be associated with the volatile nature of the EO.

Although they were not presenting all peaks (100), (110), and (200), characteristic of highly ordered SBA-15 porous structure [30], the SAXS diffractograms of bare, encapsulated, and capped samples revealed the partially ordered structure of MSPs (Figure 4a–c). This is evidenced by the presence of (100) peak. As can be noticed in Figure 4c, the signal of the latter became remarkably higher with the outer coating layer of chitosan. This could be due to the change of scattered intensity induced by a modified electronic contrast arising from the biopolymer deposition [31,32].

The DLS measurements (shown in Figure 5B) revealed a very broad size distribution of AP-EO/MSPs, within the micrometer scale. It started from almost 130 nm up to more than 2400 nm, and was centered at ~1000 nm. Due to the hydrophobic character of the essential oil encapsulated in mesoporous silica single particles, the latter tended to form very large aggregates, yielding such size values. The same explanation could be given to account for the size distribution of AP-EO/MSPs Chi, as illustrated in Figure 5B. It was within the 2500–7400 nm range, and centered at 4788 nm. The pore capping with the chitosan layer induced an enlargement of the AP-EO/MSPs’ particle size. It is interesting to mention that water was used for the dispersion to avoid any release of the essential oil from the porous structure when using an organic dispersing solvent.

### 2.2. Encapsulation Efficiency and EO Release

The evaporation of the non-encapsulated EO was observed mostly during the first 24 h of the experiment, reaching 68.6% weight loss (Figure 6). Then, the evaporation continued gradually up to 144 h and stabilized when it reached 76.6% until the end of the experiment. The residual weight could be explained by the presence of EO compounds with low sensitivity to the experimented evaporation temperature (23 to 28.5 °C) as well as non-volatile compounds. The obtained AP-EO/MSPs released EOs more progressively for 72 h. In fact, the AP-EO/MSPs’ weight was reduced by 29.2% during the first 24 h. Then, the weight of these particles decreased progressively along 72 h reaching up to a 41.5% loss. The residual mass is equivalent to the sum of residual EO compounds’ and the MSPs’ weights. Consequently, LE% was estimated to be 69.9%. Additionally, it is interesting to mention that the weight evolution of the AP-EO/MSPs coated with chitosan did not present a significant variation over time, showing there was no significant quantity of EO released.

Furthermore, absorbances measured within the diffusion assays showed increases of EO concentrations outside the dialysis bags when it came from both free EOs and/or AP-EO/MSPs in solution at pH 7.6. A maximum absorbance after 6 h of the experiment was recorded (Figure 7). However, in the case of the chitosan-coated MSPs, no release of EO was observed. EO diffusion continued progressively. Thus, the release percentages measured for AP-EO and AP-EO/MSPs from 72 to 168 h were 75.3% and 39.7%, respectively. Considering the non-diffusible residue of EO, it could be estimated that 100 mg of loaded MSPs contained 52.7 mg of AP-EO. Evidently, this result confirmed the adsorption of the EO in the MSPs. This encapsulation extended the EO release to 72 h compared to the free EO that had totally evaporated within 24 h. As shown, the use of chitosan coating prevented the EO from being released both in the air and in the PBS buffer.

### 2.3. Antifungal and Antimycotoxic Activity of the AP-EO Formulations

The antifungal activities in contact assays of AP-EO inhibiting the *F. avenaceum* mycelial growth were dose-dependent for both free AP-EO and AP-EO/MSPs (Figure 8). Indeed, free AP-EO showed an inhibition of 15.8% at 0.05 µL mL^−1^ and 82.9% at 0.1 µL mL^−1^ (Figure 9a). Whereas, in the case of AP-EO/MSPs, 31.1% inhibition was recorded at 0.1 mg mL^−1^ and 93.7% at 0.2 mg mL^−1^ (Figure 9b). In fact, because 52.7 % of AP-EO/MSPs’ weight was EO (Figure 6), thymol has a density value of 0.965 g/mL at 25 °C, and thymol was the main volatile compound of AP-EO (53% of volatile compound composition as shown in [14]), it can be argued that the treatment with 2 mg of AP-EO/MSPs was comparable to the treatment with 1 µL of free AP-EO, in terms of EO quantity. IC50 values were 0.067 and 0.055 µL EO mL^−1^ for AP-EO and AP-EO/MSPs, respectively. Consequently, for equivalent amounts of free EO and EO loaded in MSPs, the latter presented higher growth inhibition values. While the non-loaded MSPs (control in Figure 9b), as well as all the treatments with chitosan-coated MSPs (Figure 9c), did not affect the mycelial growth of *F. avenaceum*.

The ENNs accumulation in the culture medium was expressed as µg released into the culture medium relative to the mycelial surface covering the medium. It was significantly inhibited at rates varying from 80% to 86% by free AP-EO and by 85% to 92% by AP-EO/MSPs (Figure 9d,e) compared to the treatment without AP-EO. No significant concentration dependency effect was observed at the tested treatments. Surprisingly, the exposure to non-loaded MSPs (control in Figure 9e) compared to the test without EO (0 in Figure 6d), significantly limited ENN accumulation by 61%, while the mycelial growths were comparable. There was an inhibition of ENN synthesis. The addition of AP-EO to MSPs resulted in a significant decrease of accumulation up to 80 % of the value obtained with MSPs alone.

Moreover, coating MSPs with chitosan did not significantly change the effects of MSPs and AP-EO/MSPs on ENN accumulation (Figure 9f), despite the absence of effect on the mycelial growth rate. With AP-EO/MSPs-Chi, the decrease in ENN accumulation was improved when compared to MSPs-Chi, but not when compared to AP-EO/MSPs. Furthermore, in absolute values, lower quantities of ENNs were present in the Petri plates of the AP-EO/MSPs-Chi treatments than the AP-EO/MSPs ones due to the differences in mycelial growth (Figure 8b,c). There was an additive inhibition of ENNs synthesis.

The exposure to AP-EO volatile compounds in the fumigation assay had similar effects on the mycelial growth compared to the contact assays (Figure 10). Significant dose-dependent inhibition was observed for both free AP-EO treatment and AP-EO/MSPs treatment (Figure 11a,b). IC50 was measured to be 4.86 µL per flask. The AP-EO/MSPs presented higher percentages of inhibition compared to the equivalent amounts of the free AP-EO and an IC50 at 3.15 µL per flask. Chitosan-coated particles were not applied in this experiment since the coating prevented the release of the volatile components of AP-EO. It is considered that these particles could represent an interesting system which is preloaded with the bioactive products and coated to be sealed for further use.

Unlike the contact assays, a significant dose-dependent inhibition was observed for ENN accumulation during the fumigation assays (Figure 11c,d). In fact, an inhibition was observed of ENN production per cm^2^ of the mycelial growth by 70%, 90%, and 100% with 5 µL, 7.5 µL, and 10 µL of free AP-EO, respectively. While for the AP-EO/MSPs an inhibition was recorded of the ENN production per cm^2^ of mycelial growth by 94%, 98%, and 100% with 10 mg, 15 mg, and 20 mg per flask, respectively.

## 3. Discussion

### 3.1. Enhancement of EO Antifungal Activity by Encapsulation in MSPs

Emerging mycotoxins are raising serious concerns on the scientific community. Studies are interested in the toxic potential of these mycotoxins and the appropriate control strategies to prevent their accumulation in food crops [33,34]. Thus, EOs have been investigated as a safe, ecofriendly, renewable, and easily biodegradable option to be used to prevent both the fungal contamination and the mycotoxin accumulation in food commodities [15].

Various studies have investigated the antifungal effect of several EOs against *F. avenaceum* using different experimental methods such as the agar dilution method, the disk method, or the fumigation method. EOs from more than 45 species have been shown by different authors to exert antifungal effects [14]. For example, Hanana et al. [35] reported that *Origanum vulgare* EO inhibited 77.4% of *F. avenaceum* growth when tested with the agar dilution method, and showed that thymol (29.6%) and p-Cymene (29.4%) were the major compounds when the EO chemical profile was assessed. Furthermore, in the same study, *Thymus capitatus* EO with carvacrol as a major compound (69.15%) inhibited 89% of *F. avenaceum* mycelial growth. In our previous research, the antifungal effect of eight different EOs (*A. pusilla, T. capitatus, Carumcarvi, O. vulgare, Myrtuscommunis, Artemisia absintum, Mentha spicata*, and *Schinus terbenthifolius*) was investigated, showing that the two EOs *A. pusilla* and *T. capitatus* demonstrated the highest inhibition level of *F. avenaceum* mycelial growth [14]. When using the fumigation assays with *A. pusilla* and *T. capitatus* EOs over a long incubation time, phases of almost total inhibition were observed for several days, but finally the mycelia growth capacity was recovered after 10 to 24 days [13,14]. Thus, a stabilization system of the EO was needed to extend its release.

Encapsulation is one of the most promising techniques to protect the EOs and control their delivery. Silica mesoporous particles (MSPs) are known to be a stable platform with homogeneous porosity that provide a high surface area and a high loading capacity. Furthermore, MSPs’ surface enables a high range of functionalization with a variety of molecules in different regions of the particles. This property makes MSPs highly versatile and able to perform specialized tasks [20,24].

The EO-loading encapsulation percentages in MSPs vary in different studies. In the present work, LE% was 42.2% for AP-EO. This value was confirmed by thermogravimetric analysis (45%). It was higher than the LE% reported by Sattary et al. [24], which were 33.75% and 26.9% for *Cymbopogon citratus* and *Syzygium aromaticum* EO, respectively. This is in contrast with Ebadollahi et al. [23] who reported LE% of 89.1% and 85.5% for *Thymus eriocalyx* and *Thymus kotschyanus* EO, respectively. In fact, the methods used for preparing the EO encapsulation in MSPs varied between the authors. However, we measured 69.9% of LE% for *T. capitatus* EO (data not shown), showing that the nature of the EO may also affect the capacity of encapsulation into MSPs. Specifically, the encapsulation efficiency, surface area, and pore size are controlled by the type of oil and surfactant. With the cationic surfactant we used, CTAB, the positive charge neutralized the negatively charged silica particles and generated chain–chain interactions by surfactant adsorption, resulting in an increment of pore size and surface area [36]. Janatova et al. [21] tested several EO compounds encapsulated individually in equivalent amounts against *Aspergillus niger* and showed that the compounds’ release depended on its chemical nature. The release is also dependent on the pore size. The low pore size (3.1 nm) we obtained explains the low kinetics of release of AP-EO from AP-EO/MSPs [37].

The antifungal activity of the AP-EO/MSPs against *F. avenaceum* showed a higher mycelial growth inhibition compared to the free AP-EO in contact assays when the products were included in an agar medium. Furthermore, measuring only the effect of volatile compounds released from a source outside of the agar medium, there also was an observed improvement in the antifungal activity with AP-EO/MSPs at equivalent concentrations of available EO compared to free AP-EO. This result is similar to the previous finding of Ebadollahi et al. [23] that reported a better activity against two-spotted spider mites of *T. eriocalyx* and *T. kotschyanus* EOs loaded in MCM-41 compared to the free EOs. Furthermore, the encapsulation did not only increase the effect of the EO, but also extended its release into the fungal environment by 13 days. It was reported an eight-fold increase of the antifungal activity against *A. niger* of MSP-encapsulated thymol compared to the non-encapsulated compound and its diffusion was considerably lowered [21]. In a recent study, Sattary et al. [24] reported a three-fold increase of lemongrass and clove EOs activity against *Gaeumannomyces graminis* var. *tritici*, a causal agent of take-all disease of wheat, when encapsulated into MSPs, which improved their stability and solubility. In fact, the MSPs’ surface increased the solubility of the EO in water by reducing its hydrophobicity, allowing the EO to disturb the membrane integrity and destroy the fungal cells [21,24]. Thus, a lower EO dosage will be necessary to achieve the same antifungal activity as estimated by free EOs. This is an advantage due to the limits in the production of EOs.

In the present study, the non-loaded MSPs used as control did not affect the mycelial growth, but it was reported that MSPs could act as a phytostimulant on wheat, accelerating the germination of seeds, increasing the development of roots, and other plant growth parameters [24,38]. Following root uptake, MSPs were localized in chloroplasts [38]. MSPs applied in soil improved the growth of maize [39]. It had also been reported that non-porous silica nanoparticles increased maize disease resistance against *Aspergillus niger* and *Fusarium oxysporum* compared with a treatment of bulk silica [40]. Combining the antifungal activity of EO/MSPs observed in the present study and the effect of MSPs on the plant growth promotion as well as the stimulation of plant mechanism against fungi, MSPs have shown a good potential for protecting wheat and maize against infection by *F. avenaceum*.

### 3.2. Enhancement of EO Antimycotoxin Activity by Encapsulation in MSPs

Very few studies have considered the effect of the EO exposition on ENN production. In our previous work, it was observed that all the tested EOs coming from eight different plants significantly inhibited the production and accumulation of ENNs in an agar medium and the IC50 for the reduction of ENNs accumulation was estimated to be 0.10 to 0.05 µL mL^−1^ in contact assays for AP-EO [14]. The inhibition of ENN production and accumulation were confirmed in the present study. However, despite the choice of tested concentrations on the average mycelial growth IC50, a strong effect was observed with 80% inhibition at the lowest concentration. Consequently, no significant additive effect of encapsulation in MSPs was observed in terms of inhibition of synthesis per unit of biomass. However, because it had the highest antifungal activity, it was observed that 1/2 to 1/5 of the quantity of ENNs accumulated in the plates of agar medium with AP-EO/MSPs treatments compared to AP-EO treatments at equivalent concentrations. This observation is demonstrating for the first time the benefits of using encapsulated EOs in MSPs for limiting the contamination of a matrix by mycotoxins produced by a *Fusarium* species.

In the present study, and to the best of our knowledge, it was also the first time a significant effect of the non-loaded MSPs on the ENN production by a *Fusarium* species was observed. The mechanisms were not studied here, but interference with the antioxidant regulation was suspected. Sun et al. [41] have shown that MSPs penetrated the plant cell wall of wheat and lupin seedlings, and moved through the walls and intercellular spaces. A recent review proposed an understanding that most crop improvements with MSPs can be explained by MSPs’ intricate correspondence with phytohormones, antioxidants, and signaling molecules [42]. In particular, MSPs can upregulate the antioxidant system. Similarly, they could penetrate the fungal cell and affect the regulation mechanisms of ENN synthesis. Furthermore, antioxidant properties of several compounds are assumed to play a primary role in antifungal and mycotoxin inhibitory activities [43]. Further studies would be necessary to identify the mechanisms involved in the ENN synthesis-inhibiting effect of MSPs.

### 3.3. Chitosan Coating of MSPs

For further control of the release of the encapsulated EO and to obtain an encapsulation protecting its properties and control its release over time, a chitosan coating was used. Chitosan is a cationic polymer comprising β-(1–4) linked D-glucosamine and N-acetyl-D-glucosamine units, produced by alkaline deacetylation of chitin, which is the most widely found polysaccharide in nature after cellulose and a main component of the exoskeleton of crustaceans as well as fungal cell walls [44]. Therefore, it represented an excellent candidate as nanoencapsulation material to explore its previously reported antifungal activity [16]. Indeed, when exposed to chitosan at 0.98 aw, *F. graminearum* mycelial growth decreased significantly [27].

Various studies reported the effect of the encapsulation of multiple EOs and their compounds on the inhibition of mycotoxin production. This inhibition varied from 59% to 100% of different mycotoxins (*Fusarium* and non-*Fusarium* toxins) such as DON, 3ADON, 15ADON, fumonisins, and aflatoxins [28,45,46,47,48]. Furthermore, it was reported there was a two-fold increase of the antifungal activity of encapsulated clove EO against *A. niger* [25]. However, to the best of our knowledge, the effect of such encapsulation on ENN production by *Fusarium* species was not reported elsewhere. In the contact assays, the chitosan-coated MSPs with or without EO had a significant inhibition activity on ENN production by *F. avenaceum.* Additionally, the combination of mesoporous silica encapsulation of EOs and a chitosan coating to improve the stabilization and extend the release of the EO, has not been reported. Our results show that the chitosan coating can prevent the release of AP-EO from MSPs until the time of use. It is a solution for long-term preservation of the formulated AP-EO/MSPs. When used in contact with the fungi, AP-EO/MSPs-Chi would be an excellent biostimulant for the plant, and will also inhibit the production of ENNs by *F. avenaceum*. Chitosan would likely be altered by its extracellular N acetylglucosaminases and chitanases, releasing the active AP-EO/MSPs that will limit both the growth of the fungi and lengthen its antimycotoxic properties. It was reported that degradation of 50% of chitosan was obtained in complex soil (silty soil) after 10 days, and it was totally degraded after 30 days at 25 °C due to the presence of actinobacteria. In fact, chitosan can be affected by the eukaryotic cells’ lysozymes that recognize the N-acetylglucosamine sequence, but also with several diluted acids such as formic acid with hydrogen peroxide [49] that could be improved using a sonication procedure [50]. Apart from the chemical and enzymatic processes for chitosan degradation, physical methods have also been used such as ultrasound, electron-beam plasma, solution plasma, cavitation, mechanical milling, microwaves, or photo-irradiation. All of the procedures are expensive, energy consuming, and with a relative efficiency depending mainly on the initial molecular weight of the chitosan. Therefore, a chitosan coating is an interesting use for this wonder biomaterial, but its use is slightly dependent on its natural biodegradability potential for an efficient use in agriculture. Future investigation is needed to select an appropriate chitosan preparation method leading to chitosan with both stronger bioactivities [51] and progressive biodegradation susceptibility for releasing the coated active components. Providing a reproducible procedure to destabilize this coating just before the application in order to reactivate the potential of the AP-EO/MSPs could also be an alternative.

## 4. Materials and Methods

### 4.1. Mesoporous Silica Particles Synthesis

MCM-41 (Mobil Composition of Matter No. 41) has a hexagonal array of uniform pores and channels and it belongs to a family of mesoporous materials known as M41S [52]. MCM-41 silica particles were synthetized using tetraethyl orthosilicate TEOS (Si(OCH_2_CH_3_)_4_) as precursor in acidic environment based on the sol–gel method. A mixture was prepared using 16 mL of H_2_O, 20 mL of hydrochloric acid (HCl), 0.2 g of CTAB as surfactant, 10 mL of absolute ethanol (EtOH), and 1.2 g of P123, and stirred at 40 °C for 45 min. Next, 4 mL of TEOS was added and stirred at 40 °C for 45 min. The resulting solution was heated at 70 °C for 16 h and a white suspension was formed. Then, a decantation using a Soxhlet apparatus using ethanol as solvent for 48 h was performed to collect the powder that was left to dry overnight at 70 °C leading to MSPs.

### 4.2. Nitrogen Physisorption Measurements at 77K

The characterization of textural and morphological properties of the synthesized MSPs were carried out using a Belsorp-MAX device (Japan bel Co, Ltd.,Tokyo, Japan). It is worth noting that prior to the N2 physisorption measurements at 77K, the powdered sample was degassed at 100 °C for 3 h in order to eliminate the adsorbed species remaining in the porous network. The specific surface area (SSA) was determined using the Brunauer–Emmett–Teller (BET) method [53] and the pore size distribution (PSD) was evaluated using Barrett–Joyner–Halenda (BJH) analysis [54].

### 4.3. XPS

The MSP sample was characterized using a K Alpha apparatus (Thermo, Waltham, MA, USA). A monochromated Al Kα source was used (hν = 1486.6 eV, spotsize = 400 µm). A flood gun was used to compensate for the static charge built on the surface. The pass energy was set to 200 eV to record the survey region, and 80 eV for the high-resolution spectra. These conditions ensured the ability to obtain high counts without compromising the spectral resolution. The manufacturer’s sensitivity factors were used to determine the composition.

### 4.4. Thermogravimetric Analysis (TGA)

TG analyses of both bare and EO-loaded MSPs were conducted using a Setaram instrument (set sys evolution 16 model) (KEP Technology, Sophia Antipolis, France). Practically, the obtained powder was heated up from room temperature to 800 °C at a linear heating rate of 10 °C min^−1^ under air flow.

### 4.5. Small Angle X-ray Scattering (SAXS)

The SAXS measurements were carried out using an Empyrean diffractometer equipped with a copper tube and multichannel detector.

### 4.6. Dynamic Light Scattering (DLS)

The Dynamic Light Scattering (DLS) measurements were performed using a Malvern Nano ZS apparatus model ZEN3600. Prior to analysis, powdered samples were dispersed in distilled water.

### 4.7. Essential Oil Encapsulation and Chitosan Coating

*Ammoides pusilla* essential oil (AP-EO) was obtained as described previously by Chakroun et al. [13,14]. The impregnation method was used for loading the EO into MSPs. First, 200 µL of EO was diluted in 400 µL of hydroalcoholic solution (70/30, *v*/*v*). The EO solution was added drop by drop to 200 mg of MSPs. The solvent was evaporated at room temperature and the loaded particles (AP-EO/MSPs) were collected.

The chitosan coating was performed as follows: first, 200 mg of chitosan was dissolved in 50 mL of solution of acetic acid (10%). Then, 200 mg of collected loaded particles were added to 40 mL of the prepared chitosan solution and stirred for 36 h at ambient temperature. The resulted product was collected after 2 cycles of centrifugation (1200× *g*)/rinsing acetic acid solution at 10%. The powder of AP-EO/MSPs coated with chitosan (AP-EO/MSPs-Chi) was obtained after final evaporation at ambient temperature.

### 4.8. In Vitro Release of EO in Liquid and Evaporation

The evaporation of AP-EO from the loaded MSPs was monitored using either weight loss of 12.5 µL of EO or of 20 to 24 µg of EO-loaded MSPs placed at room temperature in small tared weighing dishes. The weights were recorded after 6 and 24 h, and daily for 9 days. The mean room temperature was 25.2 ± 2.5 °C. Controls containing non-loaded MSPs were used for measuring variations in weight due to the changes in air relative humidity, and to correct the measured values of EO-loaded MSPs using absorption/release factors. The mean relative humidity throughout the incubation was 62 ± 10%. The loaded EO percentage (LE%) was calculated using the following equation:LE% = mass of loaded EO/Mass of loaded particles × 100

To measure the EO release from the MSPs, a dialysis method was used as described previously by Qiu et al. [55] with modifications. Briefly, 12.5 µL of AP-EO or 25 mg of AP-EO/MSPs or 25 mg of AP-EO/MSP-Chi were dispersed in 4 mL of PBS buffer (pH 7.6), and loaded in dialysis tubing cellulose membranes closed at both ends. The dialysis bags were placed in 45 mL of PBS containing 20% EtOH 95 and stirred gently at 25 °C. Samples of 3 mL of the solution were collected at 0, 6, 23, 30, 48, 72, 96, and 168 h and replaced by 3 mL of the PBS buffer + EtOH solution. The concentrations of EO components released through the membranes were estimated by changes in absorbance at 275 nm measured in the collected medium using a UV/Vis spectrophotometer. Since at each sampling time, a percentage of 6% (3 mL/45 mL) of the released EO was removed from the external compartment, a dilution factor was introduced with the addition of fresh PBS/EtOH solution, and a correction factor was applied to the absorbances as follows:A(n) = A/0.933 + (A(n − 1) ∗ 0.067)
where A is the measured absorbance, A(n) is the corrected absorbance value at the sampling time n, and A(n − 1) is the corrected absorbance measured at the previous sampling time.

### 4.9. Antifungal Activity of AP-EO Loaded in MSPs against F. avenaceum

The evaluation of the antifungal and antimycotoxigenic activity of AP-EO and its nanoproducts was performed both by contact assays using the agar dilution method and fumigation assays using the strain of *F. avenaceum* I496 as described previously by Chakroun et al. [13,14]. FDM-agar medium, per liter of distilled water, contained the following: 12.5 g of glucose, 4.25 g of NaNO_3_, 5 g of NaCl, 2.5 g of MgSO_4_7H_2_O, 1.36 g of KH_2_PO_4_, 0.01 g of FeSO_4_ 7H_2_O, 0.0029 g of ZnSO_4_7H_2_O, and 15 g of agar. The pH of the FDM medium was 7.2. For the contact assays, FDM-agar medium was cooled at 50 °C after sterilization. Solutions of AP-EO in EtOH were prepared and added in sterile conditions with constant stirring to obtain final concentrations as follows: 0, 0.05, 0.075, and 0.1 µL mL^−1^ in 2% of EtOH. The loaded particles and the chitosan particles were diluted in the FDM-agar medium before pouring in Petri plates to obtain final concentrations of 0, 0.1, 0.15, and 0.2 mg mL^−1^. A volume of 10 mL supplemented culture media and control with no supplementation were poured in 90 mm Petri plates. Six Petri dishes were prepared for each treatment and inoculated at one point with 10 µL of spore suspension of *F. avenaceum* (adjusted at 10^6^ spores per mL). Each set of treatment was incubated in different hermetically closed jars at 25 °C for 10 days. Straight linear regression equations defining the relationship between the decimal logarithms of EO concentrations and either the growth index or the area of mycelium covering the plates at the end of the experiment (10 days) were used to calculate the IC50. On day 10, the plates were conserved at −20 °C until ENN analysis.

The evaluation of the effect of the volatile compounds of AP-EO and its nanoproducts was performed in fumigation assays. Two perpendicular length scales were marked on each 90 mm Petri plate in which 10 mL of FDM-agar medium were poured. After the medium solidification, 10 µL of *F. avenaceum* spore suspension (10^6^ spores per mL) was inoculated at one point in the center of the dishes. The required amounts of *A. pusilla* EO and its nanoproducts were added into 1 cm^2^ of sterile filter paper put in the center of open 55 mm Petri dishes. The tested AP-EO amounts were: 0, 5, 7.5, and 1.0 µL; AP-EO/MSPs amounts were 0, 10, 15, and 20 mg. For each treatment, three inoculated plates kept without lids and the respective amounts of the tested product were put in a hermetically closed glass jar. The jars were incubated horizontally at 25 °C for 10 or 24 days. The mycelial growth was evaluated after 10 days and 24 days of incubation through the marked length scale without disturbing the fungi incubation. Then, mycotoxins were extracted after both 10 and 24 days of incubation.

### 4.10. Enniatins Quantification

The content of the Petri plate (agar medium and mycelium) was mixed in 35 mL of ethyl acetate (VWR, Rosny-sous-Bois, France) for 15 min at room temperature with rotating end-over at 250 rpm. After filtration on n°4 Whatman filter paper, 5 mL was evaporated to dryness at 45 °C under gentle nitrogen flow. The dried sample was dissolved in 200 µL of methanol/water (1:1, *v*/*v*) and filtered on 0.2 µm filters before analysis. Quantification of ENNs was performed on a Shimadzu Prominence UPLC chain, equipped with a two-pump LC-30 AD, a degasser DGU-20A5R, an auto sampler SIL-30 AC, and a DAD detector SPD-M20A (Shimadzu Scientific Instruments, Noisiel, France). Separation of 5 µL of extract was achieved on a Kinetex 2,6U XB-C18-100 Å column (150 × 4.6 mm; 2.6 µm) (Phenomenex, Le Pecq, France) maintained at 45 °C. An elution gradient of acetonitrile in water was used with a constant flow at 1.4 mL min^−1^: 30% acetonitrile for 2.5 min, 30–99% acetonitrile in 5 min, 99% acetonitrile kept for 3.5 min, followed by a post-run equilibration with 30% acetonitrile for 2.5 min. LC-grade methanol and acetonitrile were purchased from VWR (Fontenay-Sous-Bois, France). Absorbance spectra were recorded from 190 to 450 nm and peak areas were measured at 205 nm. External calibration with standard solutions of ENNA, A1, B, and B1 (Sigma-Aldrich, St. Quentin Fallavier, France), allowed quantifications between 1 and 100 µg mL^−1^. The limit of quantification (LOQ) was 1 µg mL^−1^.

### 4.11. Statistical Analysis

Mycelial growths and ENN concentrations were analyzed by ANOVA and Duncan’s multiple range tests using SAS Software (Statistical Analysis System, version 9, Cary, NC, USA). Differences were considered at a significant level of 95% (*p* < 0.05).

## 5. Conclusions

Encapsulation of essential oils into MSPs with narrow slit pores and an interesting specific surface area had enhanced its antifungal activity against *F. avenaceum* and its ENN production when evaluated in vitro both in contact and fumigation assays. This concept has delayed the release of the EO in both cases, and the empowered activity was significantly higher when volatile components effects were tested. MSPs are generally believed to have low toxicity that is primarily influenced by physicochemical features, such as diameter size, morphology, surface charge, and functionalized groups, that broaden their applicability, but a variety of critical factors influence the toxicity assessment of MSPs [56]. Before using MSPs loaded with EO as biofungicides, the toxicity of the specific product obtained has to be evaluated. Coating with chitosan was proven to be a method for favoring long-term storage of the EO-loaded MSPs if chitosan is progressively degraded on plants. It should also limit the toxicity of MSPs delivered in fields.

Thus, this study presented a promising encapsulation system that could be developed for the potentializing of *A. pussila* EOs as a biofungicide active against *Fusarium* species. This could also be applied with different EOs such as *T. capitatus* EO or other toxigenic fungi belonging to *Aspergillus*, *Penicillium*, or *Alternaria* genus.

## Figures and Tables

**Figure 1 molecules-28-03194-f001:**
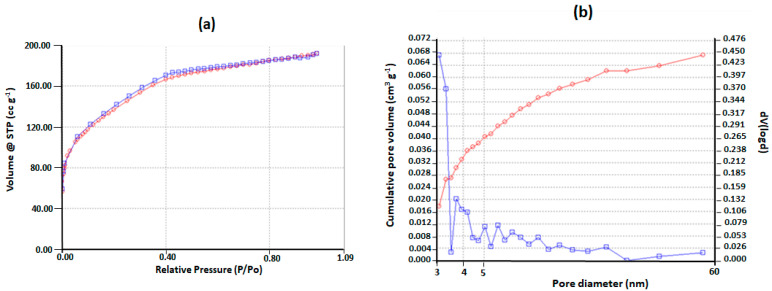
(**a**) Nitrogen adsorption (red line)—desorption (blue line) isotherms, (**b**) pore size distribution of empty mesoporous silica particles (MSPs) (blue line) and cumulative pore volume (red line).

**Figure 2 molecules-28-03194-f002:**
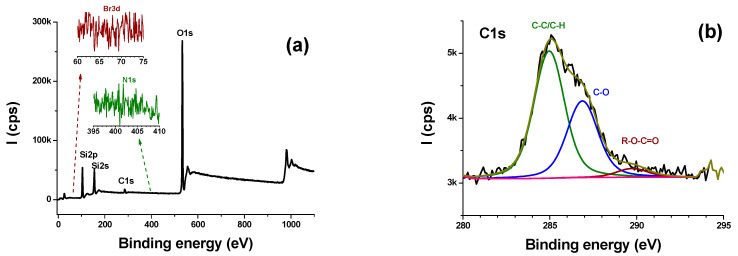
Survey (**a**), and peak-fitted C1s (**b**) spectra of MSPs. N1s (green) and Br3d (red) narrow regions are shown in inset of (**a**).

**Figure 3 molecules-28-03194-f003:**
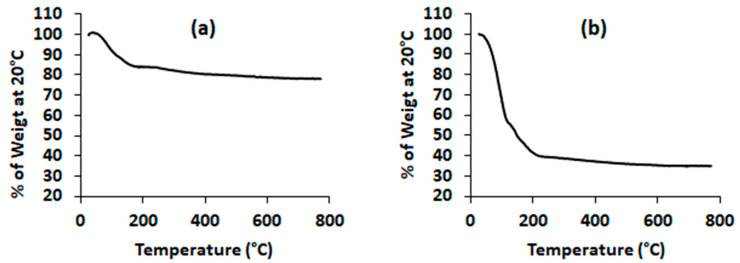
Thermogravimetric analysis (TGA) of MSPs: (**a**) empty mesoporous silica particles, (**b**) AP-EO/MSPs, mesoporous silica particles loaded with *A. pusilla* essential oil.

**Figure 4 molecules-28-03194-f004:**
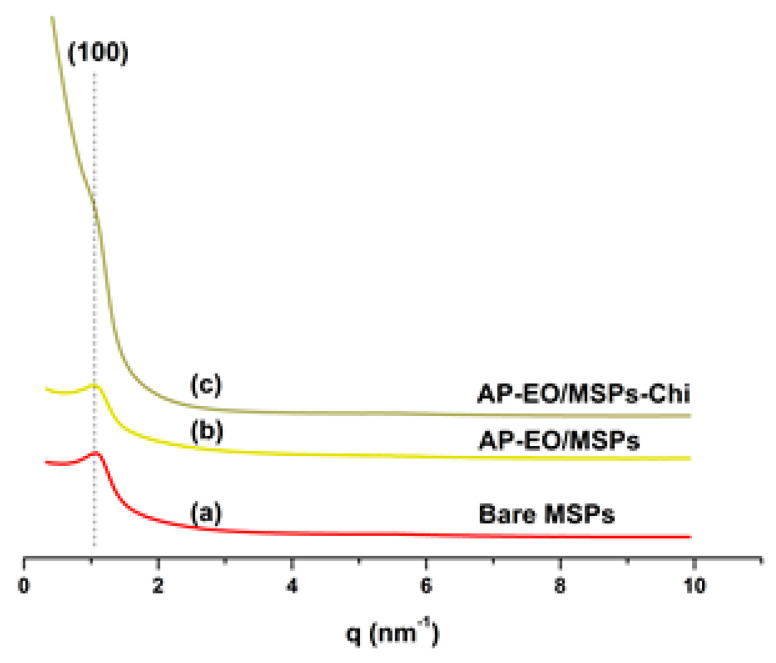
SAXS patterns of (**a**) mesoporous silica particles (MSPs), (**b**) *A. pusila* essential oil encapsulated in MSPs (AP-EO/MSPs), and (**c**) AP-EO/MSPs coated with chitosan (AP-EO/MSPs-Chi).

**Figure 5 molecules-28-03194-f005:**
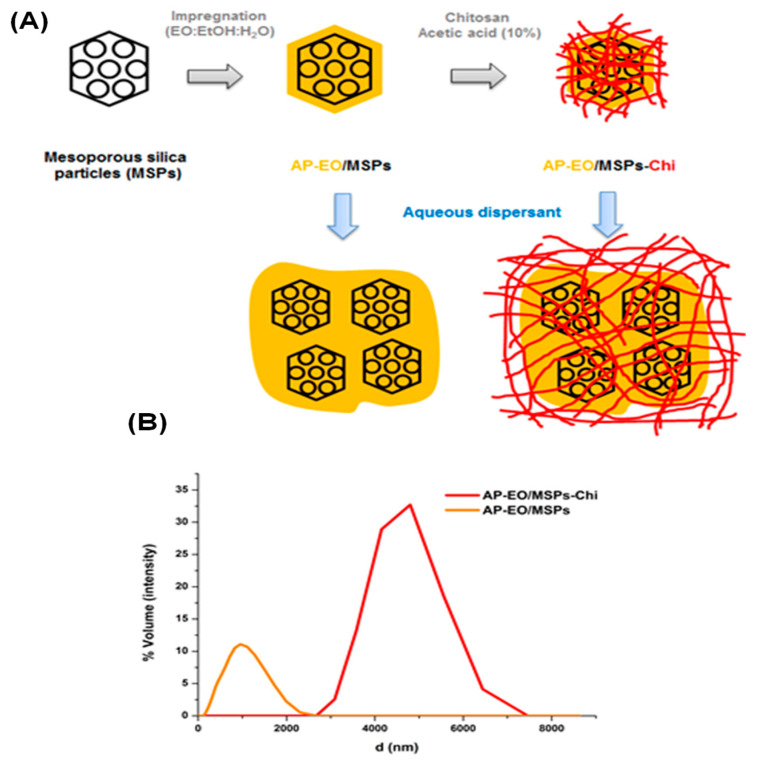
Encapsulation of essential oil in mesoporous silica. (**A**) Schematic illustration of the encapsulation process, followed by a chitosan coating of the mesoporous-silica-encapsulated oil. (**B**) DLS characterization of the mesoporous-silica-encapsulated essential oil before (orange = AP-EO/MSPs), and after its coating with chitosan (red = AP-EO/MSPs-Chi).

**Figure 6 molecules-28-03194-f006:**
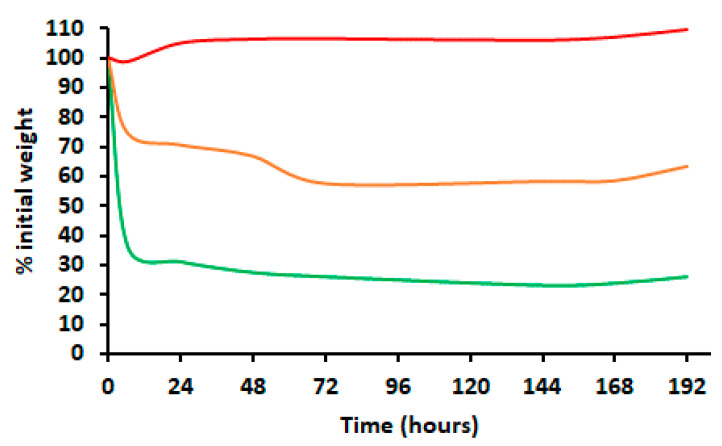
Percentages of the initial weights measured after evaporation of *A. pusilla* EO formulations at room temperature (means of 25 °C) for 192 h. Green line = AP-EO, free essential oil of *A. pusilla.* Orange line = AP-EO/MSPs, essential oil of *A. pusilla* encapsulated in mesoporous silica particles. Red line = AP-EO/MSPs-Chi, essential oil of *A. pusilla* encapsulated in mesoporous silica particles coated with chitosan.

**Figure 7 molecules-28-03194-f007:**
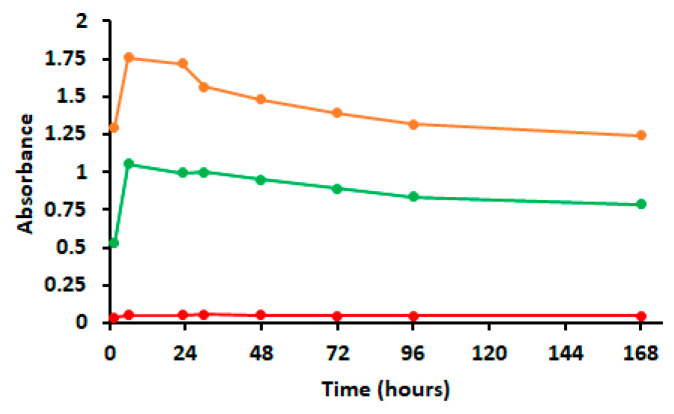
Kinetics of AP-EO release in its different formulations in PBS buffer solution. Green line = AP-EO, free essential oil of *A. pusilla*. Orange line = AP-EO/MSPs, essential oil of *A. pusilla* encapsulated in mesoporous silica particles. Red line = AP-EO/MSPs-Chi, essential oil of *A. pusilla* encapsulated in mesoporous silica particles coated with chitosan.

**Figure 8 molecules-28-03194-f008:**
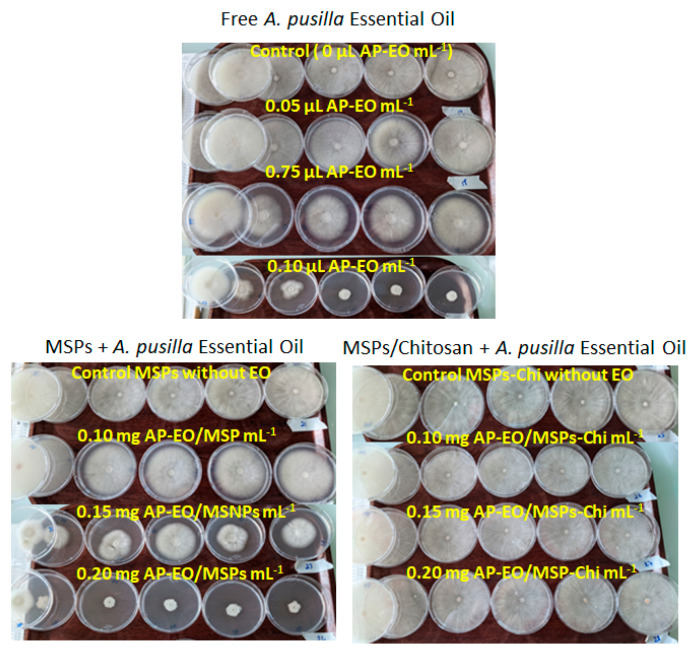
Effects of MSPs and AP-EO introduced in the culture medium on the mycelial growth of *F. avenaceum* incubated for 10 days at 25 °C.

**Figure 9 molecules-28-03194-f009:**
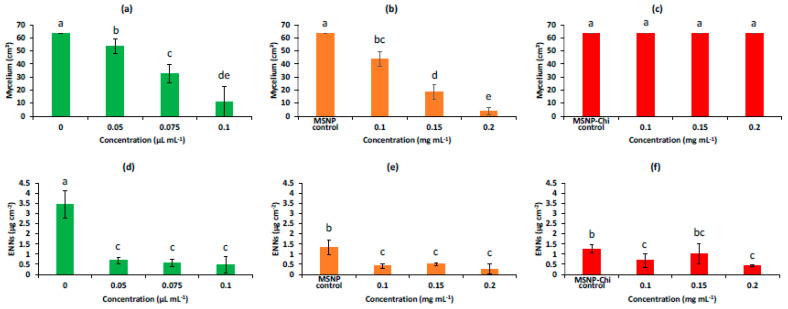
Effects in agar dilution assays of free AP-EO, (green bars, (**a**,**d**)), AP-EO/MSPs (orange bars, (**b**,**e**)), and AP-EO/MSPs-Chi (red bars, (**c**,**f**)) on mycelial growth (**a**–**c**) and on the ENN production presented as the sum of ENNA, ENNA1, ENNB, and ENNB1 (**d**–**f**) of *F. avenaceum*. Controls in (**b**,**e**) are silica particles (MSPs) at 0.1 mg g^−1^. Controls in (**c**,**f**) are MSPs coated with chitosan at 0.1 mg g^−1^. Values are means of triplicates. Error bars are ± standard deviation (SD). Means with the same lowercase letter are not significantly different (*p* < 0.05).

**Figure 10 molecules-28-03194-f010:**
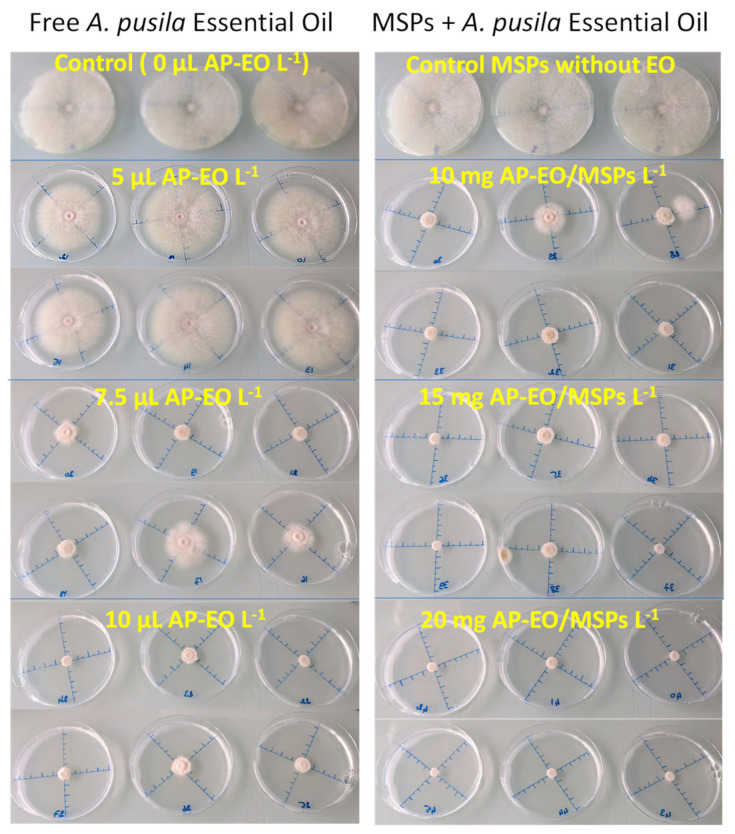
Effects of the volatile compounds released from AP-EO and AP-EO/MSPs on the mycelial growth of *F. avenaceum* incubated for 10 days at 25 °C.

**Figure 11 molecules-28-03194-f011:**
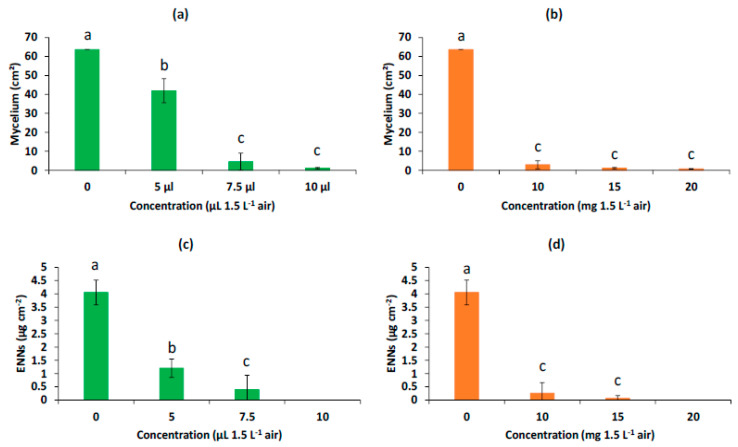
Effect of volatile components of free AP-EO (green bars, (**a**,**c**)), and of AP-EO/MSPs (red bars, (**b**,**d**)) on the mycelial growth (**a**,**b**) of *F*. *avenaceum* and on the ENN production, presented as the sum of ENNA, ENNA1, ENNB, and ENNB1 (**c**,**d**). Values are means of triplicates. Error bars are ± standard deviation. Means with the same lowercase letter are not significantly different (*p* < 0.05).

## Data Availability

The data presented in this study are available on request from the corresponding author.

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
