# Peer review of "Encapsulation of Ammoides pusila Essential Oil into Mesoporous Silica Particles for the Enhancement of Their Activity against Fusarium avenaceum and Its Enniatins Production"

_molecules, 2023, doi:10.3390/molecules28073194_

Round 1

Reviewer 1 Report

Comments:

The manuscript entitled “Encapsulation of Ammoides pusila Essential Oils into Mesoporous Silica Nanoparticles for the Enhancement of their Activity against Fusarium avenaceum and its Enniatins Production” reported an AP-EO/MSNPs for inhibition of Fusarium avenaceum. The inhibitory effect of Ammoides pusilla essential oil (AP-EO) was enhanced by coating AP-EO with mesoporous silica nanoparticles (MSNPs). AP-EO/MSNPs can effectively inhibit the growth of Fusarium avenaceum and its enniatins production. However, the antifungal mechanism was not investigated. The preparation and characterization of the material and the antifungal experiments were not systematic enough. So, I do not recommend accepting this paper right now. The manuscript needs to be resubmitted after supplementary some necessary experimental data to support the conclusion.

Major problems:

1. The preparation and characterization data of MSNPs, AP-EO/MSNPs and AP-EO/MSNP-Chi materials are not enough. To demonstrate the successful preparation of the material, the authors need to supplement the corresponding data, such as XRD, TEM/SEM, EDX element distribution, FTIR, etc.

2. There is no data on the antifungal mechanisms of AP-EO and AP-EO/MSNPs in this manuscript.

3. Figure 4, AP-EO in AP-EO/MSNPs easily evaporates at 25℃. Does this mean that AP-EO/MSNPs is unstable at 25℃? How to inhibit evaporation? How does evaporation have an impact on the application?

4. Does pH affect the release of AP-EO in AP-EO/MSNPs?

5. To support the conclusion, the authors should supplement all photographic images of Fusarium avenaceum treated with AP-EO, AP-EO/MSNPs and AP-EO/MSNP-Chi in the contact assays and fumigation assays.

6. The authors should supplement the preparation process diagram of AP-EO/MSNPs and AP-EO/MSNP-Chi.

7. There is a misspelling of words in the title, the word “Nanoparticules” needs to be corrected.

8. All of the pictures have no legends, please add them. The quality of all pictures needs to be improved.

9. In the manuscript, the use of the word, AP-EO/MSNP or AP-EO/MSNPs, should be uniform.

Author Response

We thank you for your comments.

We have performed additional analysis on the particles used in our work and followed your suggestions. The changes are in red color in the revised manuscript. For our point-by-point response, please see the attachment.

Reviewer 2 Report

The authors present very nice art of work in the manuscript, but at this point, the article needed to be improved, suggestions below,

1.       Nanoparticules! it should be nanoparticles! In the title and in the abstract it is nanoparticules, in the whole manuscript its nanoparticle, please correct it/clarify it.

2.       Electron Microscopic image is necessary for the particle before and after encapsulation with the diameter measurement, or at least the final product size distribution by TEM or the morphology of the encapsulated particles with SEM.

3.        The size of the particle should be measured before and after the encapsulation by DLS (Zetasizer) method.

4.       Turbidimetric growth curve analysis with time is necessary to evaluate the dose and time ratio with the growth of the fungus is necessary to evaluate the effectivity of the particles.

5.       The anti-test performed in the agar medium plate, then where is the plate photo and the diameter of the effected fungal colonies on the plate should be measured and included in the study.

Author Response

(The authors gave the same response as above.)

Round 2

Reviewer 1 Report

Although the authors had basically answered the reviewer's questions, please the authors  answered the reviewer's questions point-to-point.

Reviewer 2 Report

Thank you for your revision. Two points

1. For the well-shaped particles SEM or EDX-SEM is a must for their characterization, and the XRF is a plus to defend the composition present in the particle.

2. Cryo-TEM or Cryo-SEM is widely used for essential oil-coated purposes. Also here is an example published in MDPI Pharmaceuticals. https://www.mdpi.com/1424-8247/13/9/216.

After freezing or high plunge freezing, it doesn't affect the grid or plunging membranes.